

# Variation in behaviour of native prey mediates the impact of an invasive species on plankton communities

Sarah S. Hasnain[1,2] and Shelley E. Arnott[2]

[1] Department of Integrated Sciences and Mathematics, Habib University, Karachi, Sindh, Pakistan
[2] Biology Department, Queen's University, Kingston, Ontario, Canada

## ABSTRACT

Trait variation in predator populations can influence the outcome of predator-prey dynamics, with consequences for trophic dynamics and ecosystem functioning. However, the influence of prey trait variation on the impacts of predators is not well understood, especially for introduced predators where variation in prey can shape invasion outcomes. In this study, we investigated if intra-specific differences in vertical position of *Daphnia* influenced the impacts of the invasive zooplankton predator, *Bythotrephes cederströmii,* on plankton communities. Our results show that vertical position of *Daphnia* influenced *Bythotrephes* predation on smaller cladoceran species and impacts on primary production. Larger reductions in small cladoceran density and greater algal biomass were observed in mesocosms with less spatial overlap between *Daphnia* and *Bythotrephes*. These results suggest that differences in vertical position of *Daphnia* can alter the type and magnitude of *Bythotrephes* impacts in invaded systems.

# INTRODUCTION

Predation is an important mechanism structuring aquatic food webs (*Paine, 1966*; *Polis & Holt, 1992*). Predators impact prey population dynamics by directly reducing prey densities and imposing strong selective pressure on expression of prey traits, often inducing behavioural, morphological and life history changes (reviewed in *Lima, 1998*; *Tollrian & Harvell, 1999*; *Benard, 2004*), leading to indirect effects on food web structure and ecosystem function (*Paine, 1980*; *Carpenter, Kitchell & Hodgson, 1985*; *Schmitz, Křivan & Ovadia, 2004*; *Trussell, Ewanchuk & Bertness, 2003*). For example, reductions in prey density due to predation or change in prey trait expression can increase resource availability, leading to trophic cascades (*Paine, 1980*; *Carpenter, Kitchell & Hodgson, 1985*; *Trussell, Ewanchuk & Bertness, 2003*; *Schmitz, Křivan & Ovadia, 2004*). However, this research assumes that mean trait values or species identity sufficiently characterize predator–prey interactions, ignoring potential effects of intraspecific variation in both predator and prey traits on ecological communities.

Within a species, individuals can exhibit a wide range of behavioral, morphological, physiological and life history traits. For species inhabiting wide environmental gradients

Corresponding author
Sarah S. Hasnain,
sarah.hasnain@queensu.ca

or experiencing large spatial and temporal heterogeneity, differing selection pressures can shape within-species trait expression, known as phenotypic divergence (reviewed by *Crispo, 2008*; *Pfennig et al., 2010*). From an evolutionary perspective, these trait differences represent the initial stage of ecological speciation (reviewed by *Dieckmann et al., 2004*; *Pfennig et al., 2010*). Less attention, however, has been paid to the ecological consequences of trait differences between populations. Studies show that these differences can influence species abundance, community structure (*Post et al., 2008*; *Palkovacs & Post, 2009*; *Ibarra-Isassi, IT & Lessard, 2022*), and food web interactions (*Post et al., 2008*; *Benesh & Kalbe, 2016*; *Salo, Mattila & Eklöf, 2020*). For example, residence duration and foraging morphology differences between two alewife, *Alosa pseudoharengus,* populations altered prey abundances and the magnitude of trophic cascades (*Post et al., 2008*). Most studies examine ecological impacts of inter-population trait variation through predator–prey interactions (but see *Benesh & Kalbe, 2016*), focusing on the consequences of differences in predator or prey traits in simple two or three tier food webs. However, these effects have been rarely been examined in large communities where multiple species interactions are occurring across trophic levels (but see *Lenhart, Jackson & White, 2018*).

The introduction of non-native predators often results in larger negative impacts on prey communities compared to native predators (*Elton, 1958*; *Mack et al., 2000*; *Sih et al., 2010*) as prey cannot detect or deploy appropriate anti-predator responses due to absence of shared evolutionary history (*Cox & Lima, 2006*; *Banks & Dickman, 2007*; *Sih et al., 2010*; *Carthey & Banks, 2014*; *Carthey et al., 2017*). However, these studies assume that the impacts of non-native predators are uniform across the invaded range, despite the pervasiveness of intra- and inter-population variation in traits across ecological systems (*Bolnick et al., 2011*). Prey response to introduced predators depends on many factors, including the ecological novelty of the predator, the suite of anti-predator defenses available, and the degree of specialization of predator recognition templates and anti-predator defenses possible (*Carthey & Banks, 2014*). Furthermore, inter-population differences in behavioural, morphological, and life history traits may confer protection to some prey, but not others. In lakes invaded by the spiny water flea, *Bythotrephes cederströmii* (formerly *longimanus* (*Korovchinsky & Arnott, 2019*; hereafter *Bythotrephes*), a visual predator spatially restricted to the upper light-penetrating regions of the water column (*Pangle et al., 2007*; *Pangle & Peacor, 2009*; *Jokela, Arnott & Beisner, 2013*), impacts are only expected for prey populations that spatially overlap with this predator.

*Bythotrephes* is a voracious zooplankton predator (*Lehman et al, 1997*), whose introduction into North American freshwater systems has resulted in reduced biomass and species diversity of native zooplankton communities, especially for cladocerans (*Yan et al., 2001*; *Boudreau & Yan, 2004*; *Barbiero & Tuchman, 2004*; *Strecker & Arnott, 2005*; *Strecker et al., 2006*; *Kelly et al., 2013*; reviewed by *Azan, Arnott & Yan, 2015*; *Kerfoot et al., 2016*). Intense predation pressure on cladocerans, particularly *Daphnia* which are keystone grazers (*Vanderploeg, Liebig & Omair, 1993*; *Schulz & Yurista, 1998*), can induce trophic cascades leading to increased phytoplankton biomass in invaded lakes (*Strecker & Arnott, 2008*; *Walsh, Carpenter & Vander Zanden, 2016*). In some invaded lakes, *Daphnia* reside in deeper, low light-penetrating regions of the water column (*Pangle & Peacor, 2006*;

*Pangle et al., 2007*; *Jokela, Arnott & Beisner, 2011*; *Bourdeau et al., 2013*; *Hasnain & Arnott, 2019*), where *Bythotrephes* predation is reduced due to lack of visibility (*Pangle & Peacor, 2009*; *Jokela, Arnott & Beisner, 2013*). With this deep vertical position, *Daphnia* are able to avoid *Bythotrephes* predation, with potential consequences for food web functioning. Despite large variation in daytime vertical position observed across *Daphnia* populations (*De Meester, 1993*; *Tessier & Leibold, 1997*; *Boeing, Ramcharan & Riessen, 2006*), the role of vertical position of *Daphnia* in mediating *Bythotrephes* impacts in invaded systems remains unknown.

Our goal was to determine if differences in vertical position of *Daphnia* influence the effect of *Bythotrephes* on plankton communities in invaded lakes. To accomplish this, we manipulated *Bythotrephes* presence in mesocosms with three-tier food webs which were stocked from lakes with different mean *Daphnia* vertical positions and examined if the impacts of *Bythotrephes* on *Daphnia*, other cladocerans, copepod zooplankton, and algae abundance differed across mesocosms with different daytime vertical distributions. Previous studies show that *Bythotrephes* preferentially feeds on *Daphnia* as compared to other zooplankton groups such as copepods which have faster escape speed (*Schulz & Yurista, 1998*; *Pichlova-Ptacnikova & Vanderploeg, 2011*). We expected *Bythotrephes* predation on *Daphnia* to be greater in mesocosms with greater spatial overlap with *Bythotrephes, i.e.,* shallow vertical position as compared to those with reduced spatial overlap, *i.e.,* deep vertical position. Greater *Bythotrephes* predation on other cladoceran species was expected in mesocosms with less spatial overlap between *Daphnia* and *Bythotrephes*, resulting in reduced abundances of these species. We expected increases in copepod abundance in invaded mesocosms regardless of vertical position of *Daphnia* due to reduced grazing competition as a result of *Bythotrephes* preferential predation on *Daphnia* and other cladoceran grazers. We predicted that *Bythotrephes* predation on these cladoceran grazers combined with reduced *Daphnia* grazing in mesocosms with deep vertical position of *Daphnia* would result in a trophic cascade leading to increased algal biomass.

## MATERIALS & METHODS

Portions of this paper were previously published as part of a pre-print (*Hasnain & Arnott, 2022*).

### Study site and experimental design

From July 7th to August 1st 2014, we conducted a field mesocosm experiment to assess the influence of differences in vertical position of *Daphnia* on effects of *Bythotrephes* predation on plankton communities. *Bythotrephes* impacts on zooplankton community structure and algal production can occur over short periods of time comparable to the length of our experiment (*Strecker & Arnott, 2005*; *Jokela, Arnott & Beisner, 2017*; *Azan & Arnott, 2017*). Mesocosms were set up in Fletcher Lake (45.20.452′N, 78.47.798′W) Haliburton County, Ontario Canada (Table 1), where *Bythotrephes* was first detected in 2006 (*Cairns et al., 2007*). Five mil food grade polythene enclosures, 1m in diameter, 13 m deep (Filmtech Plastics, Brampton, Ontario) and filled with 10,990 L of water, were

**Table 1** Area, maximum depth, pH, average total phosphorus (TP), dissolved organic carbon (DOC) and calcium (Ca) for Fletcher Lake, Echo Lake and Bonnie Lake based on data from the Canadian Aquatic Invasive Species Network (CAISN) surveys (2008, 2011). Mesocosms were suspended in Fletcher Lake. Zooplankton were stocked from either Echo, Bonnie or combination of both lakes. *Daphnia* daytime vertical position (%) is based on data collected by *Hasnain & Arnott (2019)*. Weekly temperature measurements for mesocosms are provided in Table S1.

| Lake | Area (ha) | Maximum depth (m) | pH | TP (μg/L) | DOC (mg/L) | Ca (mg/L) | Daytime *Daphnia* vertical position (% in hypolimnion) |
|---|---|---|---|---|---|---|---|
| Fletcher | 266.28 | 23.2 | 6.12 | 5.0 | 4.6 | 2.34 | Not applicable |
| Echo | 215.6 | 11.9 | 6.46 | 9.0 | 6.6 | 2.74 | 3 |
| Bonnie | 39.3 | 21.9 | 6.78 | 5.6 | 2.3 | 2.85 | 74 |

closed at the bottom and suspended from floating wooden frames anchored in the lake. Each mesocosm was filled with water pumped from 1.5 m and filtered through an 80-μm mesh to exclude crustacean zooplankton, but allow most phytoplankton and some small rotifers and nauplii to pass through. Each mesocosm was covered with screen mesh to prevent colonization by aerial insects. Dissolved oxygen (DO) levels in each mesocosm ranged from an average of 8.9–10.6g/L at the start of the experiment. Based on data from other mesocosm experiments conducted in this region, where average DO ranged from 7.9 to 8.1 g/L over the course of 8 weeks, we did not expect DO levels to change significantly during the course of the experiment.

To assess if differences in vertical position of *Daphnia* influence *Bythotrephes* predation on plankton communities, we manipulated *Bythotrephes* presence and absence in mesocosms across a gradient of *Daphnia* vertical position. We used thermal gradients in lieu of light measurements to determine the vertical position of *Daphnia* in our mesocosms. With 4.6 mg/L of dissolved organic carbon (DOC) present in Fletcher Lake (CAISN 2011), only 1% of ultra-violet and photosynthetically active radiation (PAR) was estimated to be present at 4m (based on average summer solar radiation for south-central Ontario (*NASA Langley Atmospheric Science DataCenter, 2016*), and relationships between DOC and light attenuation (*Morris et al., 1995*). Therefore, we expected no light to be present in the hypolimnion (9–13 m) and quantified the vertical position of *Daphnia* based on their numeric proportion in this region. *Bythotrephes* predation was expected to be completely absent in the hypolimnion, where light penetration does not occur, as it is a visual predator requiring light to feed (*Jokela, Arnott & Beisner, 2013*).

After allowing phytoplankton assemblages to increase in biomass without the presence of large zooplankton (>80 μm) for three days, mesocosms were stocked with the entire zooplankton community from two uninvaded lakes; Bonnie Lake (45.17.36′N 79.06.45′W; Bracebridge Municipality) and Echo Lake (45.17.36′N, 79.06.45′W; Lake of Bays Municipality), in south-central Ontario (Table 1, *Hasnain & Arnott, 2019*). These lakes contained the same *Daphnia* species, but with contrasting vertical positions (Table 1). Each mesocosm was randomly inoculated with an ambient density of zooplankton sampled from the same volume of water from either Echo or Bonnie Lake, or half volumes for mesocosms stocked from both lakes, resulting in 16 mesocosms stocked from Echo

Lake, 16 mesocosms stocked from Bonnie Lake and eight mesocosms stocked from both lakes. This study was part of a larger experiment where an additional treatment was applied to all mesocosms three weeks after the start of the experiment. Only data from the first three weeks of the experiment was included in this study.

Half of the mesocosms were randomly assigned the invasion treatment. *Bythotrephes* individuals were collected from Fletcher Lake and Lake of Bays, Muskoka, Ontario (45°15.00′N, 79°04.00′W) using an 80 μm mesh net, and stocked at a density of 10 individuals per m³ of epilimnion volume (23 individuals per mesocosm; similar to the average density observed in invaded lakes in this region *Jokela, Arnott & Beisner, 2011*) at the beginning of the experiment. Zooplankton communities in all mesocosms were acclimated for one week prior to *Bythotrephes* addition. There were no *a priori* differences in the proportion of hypo- and non-epilimnetic (hypo- and metalimnetic) *Daphnia* between invaded and uninvaded mesocosms in week 0 (Fig. S2, Gamma GLMs, hypo: $p = 0.911$; non-epi: $p = 0.792$). *Daphnia* daytime vertical position remained stable across mesocosms throughout the duration of the experiment (see Fig. S4).

There were also significant differences in the abundance of zooplankton between Echo Lake and Bonnie Lake, which influenced initial zooplankton densities in our mesocosms. Overall zooplankton density was greater in enclosures stocked from Echo Lake as compared to Bonnie Lake (Fig. S1); log normally distributed linear model with lake origin (either Echo, Bonnie, or Both) as a predictor variable and zooplankton density as a response variable ($p < 0.0001$, Echo: $2.490 \pm 1.040$, Bonnie: $1.497 \pm 0.974$, Both: $2.910 \pm 0.861$, see Table S2 for initial densities in each mesocosm). There was no difference in zooplankton starting densities between mesocosms assigned *Bythotrephes* presence and absence treatments (Fig. S5; Absent: $1.954 \pm 1.026$, Present: $2.287 \pm 1.126$ linear model, $p > 0.05$ for all major taxonomic groups). There was very little variation in total algal densities between the mesocosms at the start of the experiment (mean = $6.438 \pm 0.110$ μg/L, Table S2).

## Sampling protocol and identification

All zooplankton samples were collected between 10 AM and 3 PM. Each mesocosm was sampled prior to the addition of *Bythotrephes* (week 0) to determine abundance and depth distribution of zooplankton and phytoplankton. Samples were also taken at the end of the study (week 3). Epi-, meta- and hypolimnetic boundaries were determined weekly using the thermal profile of Fletcher Lake using a YSI model 600 OMS V2. Zooplankton samples from each mesocosm were collected by towing a closing net with an 80 μm mesh (15 cm diameter) through each thermal layer (starting 20 cm above the enclosure bottom for the hypolimnion). Each layer was sampled in spatial order (*i.e.,* epilimnion followed by metalimnion, and then hypolimnion) to prevent disruption of any layer prior to sampling. Samples were preserved in 90% ethanol. Water samples were collected from the middle of the meta- and hypolimnion using a 2L Van Dorn sampler in weeks 0 and 3. For the epilimnion, water samples were collected from 10 cm below the surface of the mesocosm by submerging the sample container. We determined total algal biomass as well as biomass of green algae, cyanobacteria, diatoms and cryptophytes spectrophotometrically by analyzing

a well-mixed 25 ml subsample within 24 h of sample collection (BBE moldaenke Algae Lab Analyser; Schwentinental, Schleswig-Holstein, Germany).

Zooplankton were enumerated using sub-samples of a known volume and identifying all individuals within each subsample until no new species were found three sub-samples in a row. We counted a minimum of seven sub-samples for each thermal layer in each mesocosm and identified all individuals to the species level (*Ward & Whipple, 1959*; *Smith & Fernando, 1978*; *Melo & Hebert, 1994*; *Witty, 2004*; *Haney et al., 2013*). *Bosmina freyi* and *Bosmina liederi* were grouped as "*Bosmina freyi/liederi*" and *Daphnia pulex* and *Daphnia pulicaria* as "*Daphnia pulex/pulicaria*" due to morphological similarities between these species. For *Daphnia* species, only adults were enumerated. Juvenile copepods were identified as either nauplii or copepodites, without distinguishing between cyclopoid and calanoid juveniles as both have similar diets and occupy a similar trophic position (*Finlay & Roff, 2004*).

## Statistical analysis

All analyses were conducted in R v 3.2.4 (*R Core Team, 2016*) using bbmle v 1.0.17, glmmADMB v 0.7.7, fitdistrplus 1.0.7, piecewiseSEM v 2.1.2, robustbase v 0.93-3 with $\alpha = 0.05$.

Because of differences in starting densities between mesocosms stocked from Bonnie, Echo or both lakes (Table S2, Fig. S1), we standardized the change in density between weeks 0 and 3 by calculating per capita change in density for each species and functional group per mesocosm; calculated as density in week 3 divided by density in week 0, which may be sensitive to sampling and demographic effects (see Supplemental Information). We separately assessed *Bythotrephes* impacts on *Daphnia*, small (<0.85 mm) and large cladocerans (>1.0 mm) excluding *Daphnia* as well as calanoid, cyclopoid, and juvenile copepods (see Supplemental Information for details on zooplankton categorization). It is possible that the proportion of total hypolimnetic *Daphnia* may have changed during the experiment due to differences in abiotic and biotic variables between the lakes of origin and Fletcher Lake. We found no significant differences in the proportion of hypolimnetic *Daphnia* between the beginning (week 0) and the end of the experiment (week 3, see Supplemental Information for statistical assessment details).

We calculated the density of each species in each thermal layer for every mesocosm by estimating the total number of individuals present in sub samples. Species-specific density in each layer was calculated by dividing the total number of individuals by the volume sampled with a single vertical tow. To calculate the total density of a species in a mesocosm, we summed the total number of individuals in the epi- meta- and hypolimnion samples for each mesocosm and divided this by the total volume of the mesocosm that would be sampled in a single vertical tow. We could not conclusively state whether the absence of a species in our sample represented a true absence in our mesocosm as we could only sample 2% of our mesocosm volume (230L) for zooplankton and <1% for phytoplankton. We addressed this by adding minimum detection densities to all taxa assessed (see Supplemental Information).
We used piecewise structural equation modelling (piecewise SEM, *Shipley, 2016*; *Grace, 2006*; *Lefchek, 2015*) to explore causal relationships between per capita change in density of major zooplankton groups (see above), total algae biomass in week 3, *Bythotrephes* presence, and the proportion of total hypolimnetic *Daphnia* in week 0 (see Supplemental Information for SEM implementation details). We chose to assess the effect of *Daphnia* vertical position using the proportion of total hypolimnetic *Daphnia* in week 0 as it represents the *Daphnia* vertical position in mesocosms at the start of the experiment, thereby removing the possibility of any direct (*i.e.,* predation) and indirect (*i.e.,* induced response) *Bythotrephes* effects that may be present in invaded mesocosms after the start of the experiment.

Piecewise SEMs were also fit for the most common species in each zooplankton group with statistically significant causal relationships detected in the full model (Fig. 1), with *Bythotrephes* presence, and proportion of total hypolimnetic *Daphnia* in week 0 as exogenous variables, excepting paths for the most common calanoid and cyclopoid species where per capita change in copepodite density was also included. Significant causal relationships in piecewise SEMs for both common species were also re-fit with GLMs to assess if there was a significant interaction between *Daphnia* vertical position and *Bythotrephes* presence. We also refit piecewise SEMs with *Bythotrephes* presence and the proportion of hypolimnetic individuals for *Daphnia* species whose per capita change in density was identified as significantly impacted by *Bythotrephes* presence in earlier SEMs to assess if any effect of proportion of total hypolimnetic *Daphnia* observed was driven by these species. Separate piecewise SEMs were also fit for week 3 biomass of the four major algal groups; green algae, cyanobacteria, diatoms, and cryptophyta using the protocol outlined for the full model, with the nested version also confirmed using AIC.

Paths with significant causal relationships ($p < 0.05$) between per capita change in density of a zooplankton group or final algal biomass, *Bythotrephes* presence, and vertical position of *Daphnia* were re-fit with either normal or gamma distributed generalized linear models to assess the possibility of a significant interaction between the explanatory/exogenous variables. We added stocking lake as additional explanatory variable to our analysis with generalized linear models as it is possible that there are unknown differences between stocking lakes that may be underlying the observed patterns. For all GLMs, we used AICc to assess fit between gamma and log-normal distributions. Model fit was also assessed by visually examining plots of residual *versus* fitted values and square root of the standard deviance of residuals *versus* fitted values. Cook's distance was used to identify influential points (leverage >1.0). Minimum adequate models were chosen using log-likelihood ratio tests based on Crawley's (*2011*) procedure. If influential points were detected, gamma or log-normal robust GLMs were fitted using Mallows or Huber type robust estimators (*Cantoni & Ronchetti, 2001*; *Cantoni & Ronchetti, 2006*), which down weights the effect of influential points on model fit. For robust GLMs, minimum adequate models were chosen using Wald-type tests. We did not find any density dependent effects of *Bythotrephes* predation due to differences in starting densities (details about statistical methods used provided in Supplemental Information).

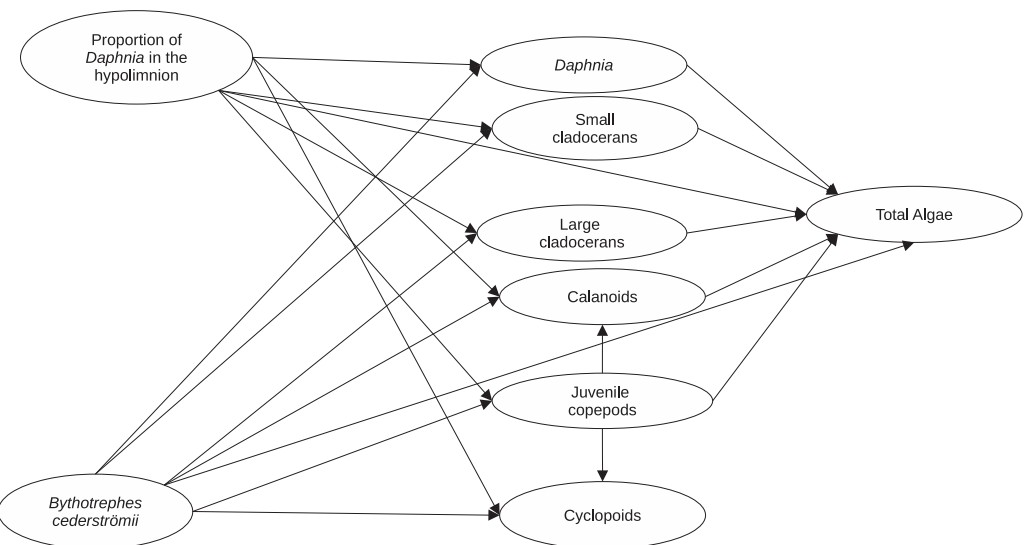

**Figure 1** **Visualization of the structural equation model (full model) used to assess the impacts of proportion of hypolimnetic *Daphnia* and *Bythotrephes cederströmii* presence on per capita change in density of major zooplankton groups and total algae.** Model assumes that all paths (represented by arrows) between per capita change density of each zooplankton group, the proportion of hypolimnetic *Daphnia* and *Bythotrephes* presence are possible.

## RESULTS

Twenty-eight zooplankton species were present, including six *Daphnia* species (*D. pulex/pulicaria, D. ambigua, D. catawba. D. mendotae, D. parvula,* and *D. dubia*) during our experiment. The most common species (>70% presence across all mesocosms) were *D. catawba* and *D. mendotae* for *Daphnia*, *B. freyi/leideri, E. tubicen,* and *E. longispina* for small cladocerans, *H. glacialis* for large cladocerans, S*kistodiapotmus oregonensis* for calanoids, and *Cyclops scutifer* for cyclopoids. Piecewise SEMs showed that per capita change in density of *Daphnia (Bythotrephes* standardized estimate (SE): −0.11, vertical position SE: 0.06), small cladocerans (*Bythotrephes* SE: −0.01, vertical position SE: 0.01), and cyclopoids (*Bythotrephes* SE: −0.11, vertical position SE: −0.28) was predicted by *Bythotrephes* presence and the proportion of total hypolimnetic *Daphnia* in week 0 (Table S5, only significant paths presented in Fig. 2). Per capita change in juvenile copepod density was only predicted by proportion of total hypolimnetic *Daphnia* (vertical position SE: 0.21, Fig. 2A, Table S5). Change in total algal biomass was predicted by per capita change in density of large cladocerans and juvenile copepods, and the proportion of hypolimnetic *Daphnia* (vertical position SE: 0.05, large cladocerans SE: 0.06, juvenile copepods SE: −0.05, Fig. 2A, Table S5).

The proportion of hypolimnetic total *Daphnia* in both invaded and uninvaded mesocosms did not change during our experiment (Uninvaded Week 0 –Week 3: $p = 0.07$, Invaded Week 0-Week 3: $p = 0.09$, $df = 76$, Table S14). The effect of *Bythotrephes* on per capita change in total *Daphnia* density was influenced by the proportion of total

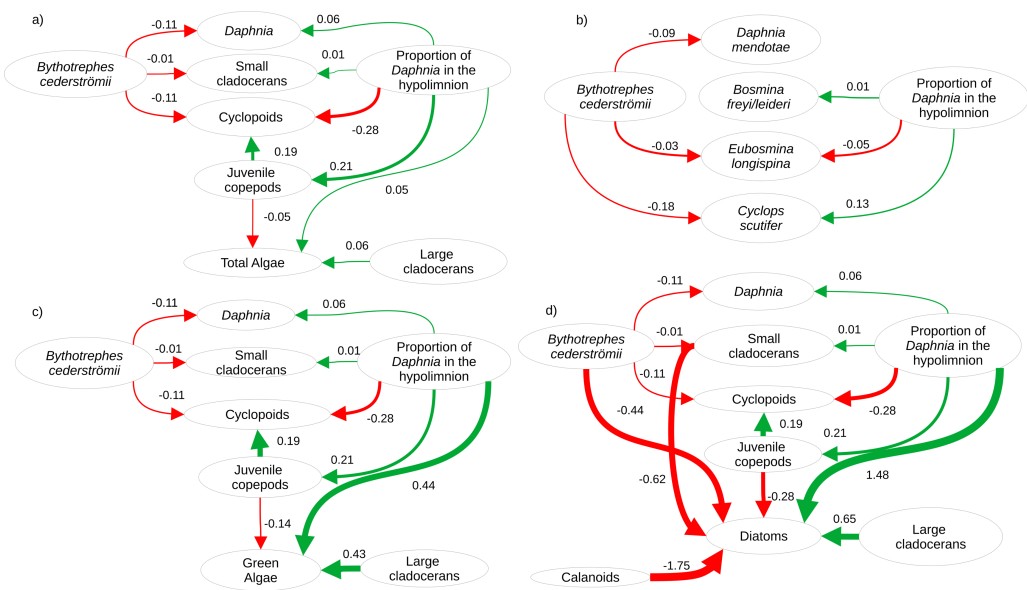

**Figure 2** **Visualization of the best structural equation model predicting per capita change in density (A) for major zooplankton groups and total algal biomass in week 3, (B) most common zooplankton species, (C) green algal biomass, and (D) diatom biomass.** Arrows represent standardized path coefficients that are statistically significant ($p < 0.05$) with standardized coefficient value. Arrow width is scaled with the size of the coefficient. Arrow colour signifies positive (green) and negative (red) path coefficients.

hypolimnetic *Daphnia* in week 0 (Gamma GLM, Fig. 3A, $p = 0.021$, $df = 36$, Table S13), with no significant effect of stocking lake ($p = 0.582$). In uninvaded mesocosms with a greater proportion of hypolimnetic *Daphnia*, per capita increase in total *Daphnia* density was larger as compared to mesocosms with fewer hypolimnetic *Daphnia*. Invaded mesocosms had a smaller per capita increase compared to uninvaded mesocosms (Fig. 3A).

Small cladocerans were mostly epilimnetic and increased in density across all mesocosms during the experiment. We observed a significant interaction between *Bythotrephes* presence and the proportion of total hypolimnetic *Daphnia* in week 0 on per capita change in small cladoceran density (piecewise SEM, Fig. 2A, Table S5). In uninvaded mesocosms, larger per capita increases in total small cladoceran density were observed in mesocosms with a greater proportion of total hypolimnetic *Daphnia* in week 0 (Gamma GLM, Fig. 3B, $p = 0.050$, $df = 36$, Table S13). Total *Daphnia* depth distribution influenced the magnitude of *Bythotrephes* impact on total small cladocerans, with a smaller per capita increase in density observed in invaded mesocosms with a greater proportion of hypolimnetic *Daphnia,* as compared to uninvaded mesocosms (Fig. 3B). We observed a significant effect of stocking lake ($p = 0.011$), with greater increase in small cladocerans observed in mesocosms stocked from Bonnie Lake as compared to those stocked from Echo Lake or both lakes (Fig. S8).

## Algal biomass
We observed a significant interaction between *Bythotrephes* presence and proportion of hypolimnetic *Daphnia* in week 0 on total algal biomass at the end of study (Gamma Robust Regression, $p = 0.0005$, $df = 314$, Table S13). Total algal biomass was greater in invaded
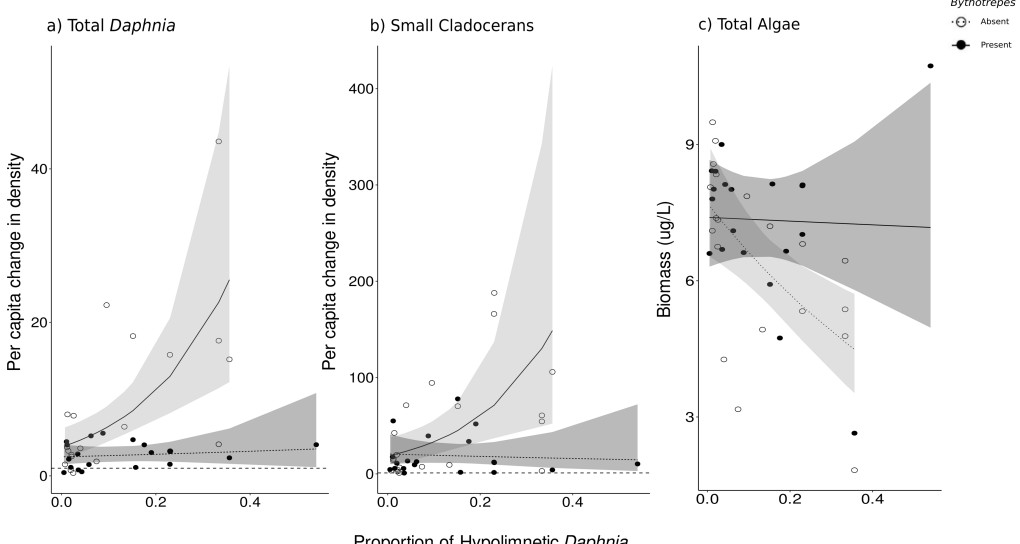

**Figure 3** **Effect of proportion of total hypolimnetic *Daphnia* in week 0 on the per capita change in (A) total *Daphnia* and (B) small cladoceran density, and (C) total algal biomass in week 3 in *Bythotrephes* absent and present mesocosms.** Values above the dashed line at 1 indicate increasing density between week 0 and 3. Shaded regions represent the 95% confidence interval estimated from the best fitting model.

mesocosms as compared to uninvaded mesocosms, with the greatest increase observed in mesocosms with greatest proportion of hypolimnetic *Daphnia* in week 0 (Fig. 3C, Table S13). Total algal biomass at the end of the study was greater in mesocosms stocked from Echo Lake as compared to those stocked from Bonnie Lake or both lakes ($p = 0.001$, Fig. S8).

Cyanobacteria were the most abundant functional group, followed by green algae, diatoms, and cryptophytes. Green algal biomass was predicted by the proportion of total *Daphnia* in the hypolimnion, and per capita change in density of large cladocerans, and juvenile copepods (Fig. 2C, vertical position SE: 0.44, Large cladocerans SE: 0.43, Juvenile copepods SE: −0.14, Table S6), and was greater in mesocosms with a greater proportion of hypolimnetic *Daphnia* in week 0 (Fig. 4E, $p = 0.010$, $df = 314$, Table S13). Green algal biomass was greater in mesocosms stocked from Bonnie Lake as compared to those stocked from Echo Lake or both lakes (Fig. S8, $p = 0.02$ $df = 37$, Table S13). Diatom biomass was predicted by *Bythotrephes* presence, the proportion of hypolimnetic *Daphnia*, and per capita change in density of small cladocerans, large cladocerans, calanoids and juvenile copepods (Fig. 2D, Table S7, *Bythotrephes* SE: −0.44, vertical position SE:1.48, Small cladoceran SE: −0.62, Large cladoceran SE: 0.65, Calanoid SE: −1.75, Juvenile copepods SE: −0.28), and was greater in mesocosms with a greater proportion of hypolimnetic *Daphnia* (Fig. 4F). We did not observe any significant interactive effect of *Bythotrephes* presence and proportion of total hypolimnetic *Daphnia* (Table S13, $p = 0.15$) or any effect of stocking lakes on diatom biomass ($p = 0.114$). There was no effect of *Bythotrephes* presence, the proportion of total hypolimnetic *Daphnia*, or per capita change in density in any zooplankton group on cyanobacteria and cryptophyta biomass (Tables S8–S9).

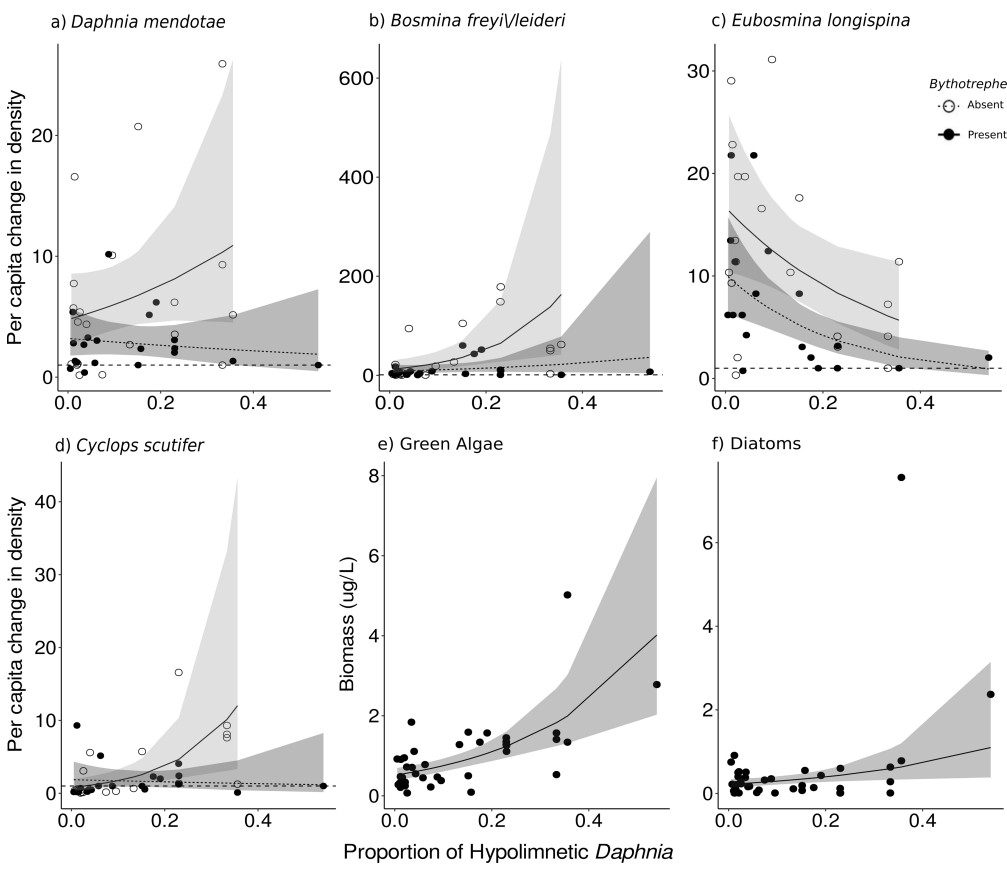

**Figure 4** Effect of proportion of total hypolimnetic *Daphnia* in week 0 on the per capita change in (A) *D. mendotae*, (B) *B. freyi/leideri*, (C) *E. longispina*, (D) *C. scutifer* density, and biomass of (E) green algae, and (F) diatoms in week 3. For green algae and diatoms, no effect of *Bythotrephes* presence was detected. Values above the dashed line at 1 indicate increasing density between week 0 and 3. Shaded regions represent the 95% confidence interval estimated from the best fitting model.

## Species-level responses

For *Daphnia mendotae*, only *Bythotrephes* presence predicted per capita change in density (piecewise SEM, Fig. 2B, Table S10, *Bythotrephes* SE: −0.09). Per capita density for *D. mendotae* was lower in invaded mesocosms as compared to uninvaded mesocosms (Fig. 4A). There was no effect of stocking lake on per capita density for *D. mendotae* ($p = 0.556$, Table S13). There was no effect of *Bythotrephes* presence or proportion of total hypolimnetic *Daphnia* on per capita change in *D. catawba* density (Table S10).

Per capita change in density in *B. freyi/leideri* was only predicted by the proportion of total hypolimnetic *Daphnia* in week 0 in our SEM analysis (Fig. 2B, Table S10, vertical position SE: 0.01), with no effect of *Bythotrephes* presence. However, we observed a significant interactive effect of *Bythotrephes* presence and *Daphnia* vertical position in our GLMs ($p = 0.008$, Table S13, Fig. 4B), with greater increase in *B. freyi/leideri* observed in uninvaded mesocosms with greater proportion of hypolimnetic *Daphnia*. Greater increase in *B. freyi/leideri* was observed in mesocosms stocked from Bonnie Lake as compared

to those stocked from Echo Lake or both lakes (Fig. S8, $p < 0.001$). Both *Bythotrephes* presence and proportion of total hypolimnetic *Daphnia* in week 0 predicted per capita change in *E. longispina* density (piecewise SEM, Fig. 2B, Table S10, *Bythotrephes* SE: −0.03, vertical position SE: −0.05). However, there was no interaction between these variables (Table S13, $p = 0.7431$). Per capita density was lesser in invaded mesocosms as compared to uninvaded mesocosms and decreased in mesocosms with a greater proportion of hypolimnetic *Daphnia* (Fig. 4C). There was no effect of stocking lake on per capita change in *E. longispina* density ($p = 0.750$). There was no effect of *Bythotrephes* presence or the proportion of total hypolimnetic *Daphnia* in week 0 on per capita change in *E. tubicen* density (piecewise SEM, Table S10).

For copepods, per capita change in *C. scutifer* (cyclopoid) density was predicted by the proportion of total hypolimnetic *Daphnia* in week 0 and *Bythotrephes* presence (piecewise SEM, Fig. 2B, Table S8, *Bythotrephes* SE: −0.18, vertical position SE: 0.13). Per capita increase in *C. scutifer* density was smaller in invaded mesocosms as compared to uninvaded mesocosms with a greater proportion of hypolimnetic *Daphnia* (Fig. 4D, Gamma GLMs, Table S13, $p = 0.02$, $df = 36$). There were no differences in per capita change in *C. scutifer* density between mesocosms stocked from Bonnie, Echo, or both lakes ($p = 0.126$, Table S13). There was no effect of *Bythotrephes* presence or proportion of total hypolimnetic *Daphnia* on per capita change in *S. oregonensis* (calanoid) density (Table S10).

## DISCUSSION

Differences in the vertical position of *Daphnia* in our mesocosms resulted in differences in the effect of *Bythotrephes* predation on the change in abundance of small cladocerans as well as individual species (*e.g.*, *C. scutifer*). Most notably, we found larger phytoplankton biomass in invaded mesocosms with a greater proportion of hypolimnetic *Daphnia* as compared to uninvaded mesocosms. This confirms our hypothesis that reduced grazing due to *Bythotrephes* predation on small cladocerans and deeper vertical position of *Daphnia* increase total algal biomass. Trophic cascades associated with *Bythotrephes* invasion are attributed to reduced *Daphnia* and cladoceran grazing due to *Bythotrephes* predation (*Strecker & Arnott, 2008*; *Walsh, Carpenter & Vander Zanden, 2016*). Our results suggest that differences in the vertical position of *Daphnia* can also contribute to trophic cascades in *Bythotrephes*- invaded systems. It is not known if differences in the vertical position of *Daphnia* can lead to different long-term community and ecosystem outcomes in *Bythotrephes*- invaded lakes. While our results show clear differences in community level impacts, these were only observed over a three-week period and we did not examine temporal dynamics over longer time scales.

Total *Daphnia* vertical position did not change during the experiment and did not influence *Bythotrephes* impact on total *Daphnia* abundance. However, *Bythotrephes* preferential predation on *Daphnia* in invaded mesocosms (*Schulz & Yurista, 1998*), resulted in smaller increases in abundances as compared to uninvaded mesocosms, matching observations from field surveys and mesocosm experiments (*Strecker & Arnott, 2005*; *Pangle & Peacor, 2006*; *Strecker & Arnott, 2008*; *Pangle & Peacor, 2009*; *Strecker & Arnott,*

*2010*; *Jokela, Arnott & Beisner, 2013*; *Azan & Arnott, 2017*). This lack of impact of vertical position of *Daphnia* on total *Daphnia* abundance in invaded mesocosms could be due to similarity between the impacts of *Bythotrephes* predation and the metabolic costs of occupying deeper colder hypolimnetic waters (*Kerfoot, 1985*; *Dawidowicz & Loose, 1992*; *Loose & Dawidowicz, 1994*; *Cole et al., 2002*; *Pangle et al., 2007*; *Pangle & Peacor, 2010*). The vertical position of the most common *Daphnia* species, *D. mendotae* and *D. catawba,* did not change during the experiment (Table S10). Increase in *D. mendotae* abundance in invaded mesocosms was less as compared to uninvaded mesocosms, suggesting *Bythotrephes* predation on epilimnetic individuals. There was no effect of *Bythotrephes* on *D. catawba* density indicating a possible preferential *Bythotrephes* predation on larger bodied *Daphnia* (*Schulz & Yurista, 1998*), *D. mendotae* over the smaller *D. catawba.*

Using per capita change in density allowed us to directly compare between *Bythotrephes* treatments regardless of differences in starting densities among our mesocosms. However, we recognize that this metric is sensitive to sampling and demographic effects. We accounted for this by adding a minimum detection density to abundances in weeks 0 and 3 for all taxa assessed prior to any analysis. This reduced the sensitivity of per capita change in density values to large changes by reducing their magnitude, especially for taxa with low abundances at the start of the experiment and high abundances at the end of the experiment, such as small cladocerans (Tables S2–S3). Low initial small cladoceran densities may be one driver for the large per capita increase we observed for this group regardless of *Bythotrephes* presence, although the observed coefficient of correlation between per capita change in small cladoceran density and initial density was not significantly different from bootstrapped values (Fig. S6).

## Indirect impacts on other zooplankton groups

Fewer small cladocerans were present in invaded mesocosms with deeper vertical position of *Daphnia* (*i.e.,* greater proportion of hypolimnetic *Daphnia*). This may be linked to *Bythotrephes* preference for large-bodied cladocerans such as *Daphnia* (*Schulz & Yurista, 1998*) in mesocosms where more *Daphnia* are epilimnetic. In mesocosms with a deeper *Daphnia*, *Bythotrephes* predation on small cladocerans led to the smaller increases observed. *Bythotrephes* predation reduces small cladoceran abundance (*Vanderploeg, Liebig & Omair, 1993*; *Yan & Pawson, 1997*; *Yan et al., 2001*; *Barbiero & Tuchman, 2004*; *Strecker et al., 2006*; *Strecker & Arnott, 2008*; *Kerfoot et al., 2016*), although this is not consistently observed (*Lehman & Cáceres, 1993*; *Barbiero & Tuchman, 2004*; *Strecker & Arnott, 2005*; *Hessen, Bakkestuen & Walseng, 2011*). The effect of *Bythotrephes* predation on small cladocerans is varied, with declines in abundance ranging from 40–126% across invaded lakes (*Vanderploeg, Liebig & Omair, 1993*; *Yan et al., 2001*; *Kerfoot et al., 2016*). Our results suggest that vertical position of *Daphnia* could be an important factor explaining this variation.

*B. freyi/leideri* density increased as the proportion of hypolimnetic *Daphnia* increased. For *E. longispina*, deeper vertical position of *Daphnia* was associated with smaller increases in density. These contrasting effects suggest that *B. freyi/leideri* in our mesocosms could be constrained by competition for grazing with *Daphnia* (*DeMott & Kerfoot, 1982*). The greater

increase in *B. freyi/leideri* density observed in mesocosms with deeper *Daphnia* vertical position may also be a result of the greater increase in density observed in mesocosms stocked from Bonnie Lake as compared to Echo Lake or both lakes. Since *Daphnia* in Bonnie Lake have a deeper vertical position, *Daphnia* vertical position in mesocosms stocked from this lake was likely deeper, thereby contributing to this observed pattern. For *E. longispina*, *Bythotrephes* predation is likely underlying the smaller increase in density observed in invaded mesocosms. This contradicts observations from literature (*Kelly et al., 2013*), where *E. longispina* abundance increased in *Bythotrephes*-invaded lakes, suggesting that the negative effect of *Bythotrephes* observed may be mediated by other factors. It is unclear what is driving the lack of increase in *E. longispina* density with deeper vertical position of *Daphnia*. A deeper vertical position of *Daphnia* may increase competition between *E. longispina* and other bosminid species.

Contrary to our prediction, *Bythotrephes* presence did not impact change in total calanoid or *S. oregonensis* density. This is surprising as negative effects of *Bythotrephes* on calanoid copepods (*Strecker & Arnott, 2005*; *Strecker et al., 2006*; *Hessen, Bakkestuen & Walseng, 2011*; *Bourdeau, Pangle & Peacor, 2011*; *Kelly et al., 2013*) have been broadly observed in the literature. Furthermore, the lack of an effect of deeper vertical position of *Daphnia* on calanoid density also contradicts expected increases due to reduced competition between *Daphnia* and calanoids (*Sommer et al., 2003*). Both *Daphnia* and calanoids primarily feed on algae, therefore less competition for algae between *Daphnia* and calanoids is expected in mesocosms with greater proportion of hypolimnetic *Daphnia,* leading to an increase per capita calanoid density. We may not have detected the impacts of these competitive interactions due to the short experiment time frame.

We observed interactive negative effects of *Bythotrephes* and the vertical position of *Daphnia* on change in total cyclopoid density. A larger increase in *C. scutifer* density was observed in uninvaded mesocosms with more hypolimnetic *Daphnia* as compared to invaded mesocosms, suggesting that *Bythotrephes* presence negatively impacts *C. scutifer* density. *Kelly et al. (2013)* observed similar *Bythotrephes* impacts on *C. scutifer* abundance in Canadian and Norwegian lakes. In contrast, no *Bythotrephes* effect was observed by *Hessen, Bakkestuen & Walseng (2011)*, while a positive effect was observed by *Walseng, Andersen & Hessen (2015)*. Although *Bythotrephes* effects on *C. scutifer* remain unresolved in observational studies, our results provide the first experimental evidence of negative *Bythotrephes* impacts on *C. scutifer* abundance. The negative effect of vertical position of *Daphnia* on total cyclopoid and *C. scutifer* density may be a consequence of indirect interactions. Deeper vertical position of *Daphnia* was linked to increased juvenile copepod density, which had a positive effect on total cyclopoid density. The negative effect of vertical position of *Daphnia* on cyclopoids may be a result of competitive interactions which were not included in our analysis.

It is unclear why *Daphnia* vertical position altered in our mesocosms as compared to their source lakes after inoculation. We expected three clearly differentiated groups of mesocosms where vertical position aligned with the lakes used for inoculation, *i.e.,* high (stocked from Echo Lake, very low proportion of hypolimnetic *Daphnia*), low (stocked from Bonnie Lake, very high proportion of hypolimnetic *Daphnia*), and mix (stocked from

both lakes, with the median proportion of hypolimnetic *Daphnia* between Echo Lake and Bonnie Lake), instead of the gradient that we observed. Daytime vertical position in *Daphnia* species is influenced by many ecological forces, including predator presence and location in the water column, predator type, food availability, exposure to ultra-violet radiation, and temperature-related metabolic costs (*Leibold, 1990*; *Boeing, Leech & Williamson, 2004*; *Williamson et al., 2011*; *Larsson & Lampert, 2012*). In addition, there is also a strong genetic component, with large intra-population variation in vertical position observed in response to predator cues (*De Meester, 1993*; *De Meester, 1996*). Similar zooplanktivorous fish species were present in all three lakes (*Ontario Ministry of Natural Resources and Forestry, 2020*), making it unlikely that these changes were driven by different predator cues. Mesocosms were filled with water from Fletcher Lake, which has similar physicochemical properties as Echo Lake and Bonnie Lake, with the exception of DOC, which was highest in Echo, followed by Fletcher, and Bonnie Lake (Fletcher: 4.6 mg/L, Bonnie: 2.3 mg/L, Echo: 6.6 mg/L). It is possible that differences in light availability and UV radiation as a consequence of differences in DOC influenced vertical position of *Daphnia*.

We also did not observe any change in the proportion of hypolimnetic *Daphnia* between week 0 and week 3 in invaded mesocosms. *Bythotrephes* can apply strong predation pressure on many *Daphnia* species (*Boudreau & Yan, 2003*; *Strecker et al., 2011*, reviewed in *Azan, Arnott & Yan, 2015*) and has been shown to induce a deeper vertical position in some *D. mendotae* populations (*Bourdeau et al., 2013*). Our observations match those of *Kiehnau & Weider (2019)*, where no differences in phototactic behaviour in response to *Bythotrephes* kairomone were observed for *D. ambigua*, *D. mendoate*, and *D. pulicaria* clones from Lake Mendota, Wisconsin prior to and after *Bythotrephes* invasion.

## Primary production and trophic cascades

*Daphnia* vertical position mediated the impacts of *Bythotrephes* on primary production by altering grazing pressure. Total algal biomass was greater in invaded mesocosms with deeper vertical position of *Daphnia* as compared to uninvaded mesocosms, likely a result of reduced small cladoceran abundance due to *Bythotrephes* predation and reduced epilimnetic grazing by *Daphnia*. Interestingly, we did not observe any effect of vertical position of *Daphnia* on total algal biomass in invaded mesocosms. For invaded mesocosms with shallower *Daphnia*, reduced grazing pressure on algae due to *Bythotrephes* preferential predation on *Daphnia* (*Schulz & Yurista, 1998*) may have resulted in the high algal biomass observed. In invaded mesocosms with deeper vertical position of *Daphnia*, *Bythotrephes* predation on smaller cladocerans in the absence of preferred *Daphnia* prey likely reduced grazing pressure on algal biomass resulting in a similar outcome. The invasion of *Bythotrephes* in some North American temperate lakes is associated with trophic cascades due to reduced grazer biomass (*Walsh, Carpenter & Vander Zanden, 2016*; *Martin, Walsh & Vander Zanden, 2022*) which has been observed in some lakes but not others (*Strecker & Arnott, 2008*). Our results suggest that in addition to the nutrient status of these lakes (*Walsh, Carpenter & Vander Zanden, 2016*), vertical position of *Daphnia* could be an important factor explaining the varied *Bythotrephes* impacts on primary production in these studies.

Total algal biomass increased across most mesocosms (Fig. S7) regardless of grazing by *Daphnia* and other cladocerans. *Daphnia* grazing was reduced in mesocosms with more hypolimnetic *Daphnia,* likely increasing green algal and diatom biomass. *Bythotrephes* presence negatively impacted diatom biomass. We also observed a negative effect of increasing small cladoceran density on diatom biomass. Apparent competition due to *Bythotrephes* preferential predation on larger-bodied *Daphnia* could lead to increased small cladoceran grazing. Furthermore, bosminids (*i.e., B. freyi/leideri, E. longispina*) were the most abundant small cladoceran taxa in our mesocosms. Their demonstrated selectivity on diatoms and green algae (*Fulton III, 1988*; *Tõnno et al., 2016*) likely resulted in the decline observed.

## CONCLUSIONS

Food web and ecosystem consequences of differences in vertical position of *Daphnia* remain largely ignored, despite literature suggesting that inter-population trait variation, especially in predator–prey interactions, may be the primary mechanism maintaining food web complexity and driving trophic cascades. This study highlights the strong influence of vertical position of *Daphnia* on interactions between zooplankton groups, ultimately affecting primary production in lake ecosystems, regardless of *Bythotrephes* presence. Our results also provide the first experimental evidence suggesting that differences in *Daphnia* depth distribution influence the impacts of *Bythotrephes* predation on other cladoceran groups, resulting in increased algal biomass. Understanding the influence of vertical position of *Daphnia* on the structure and functioning of lake ecosystems will improve our ability to predict impacts of future invasions.

## ACKNOWLEDGEMENTS

We thank the Dorset Environmental Science Centre (DESC) and its staff, especially James Rusak, Ron Ingram and Tim Field for providing logistical support during mesocosm sampling. We thank Sarah Lamb, Matthew Laird, and Shakira Azan for their field assistance.

### Funding

Funding for this project was provided by the Canadian Aquatic Invasive Species Network II (CAISN II), an Ontario Graduate Scholarship, and a graduate research award from the Muskoka Summit for the Environment. Support was also provided through the Queen's University Summer Work Experience Program (SWEP). The funders had no role in study design, data collection and analysis, decision to publish, or preparation of the manuscript.

### Grant Disclosures

The following grant information was disclosed by the authors:
Canadian Aquatic Invasive Species Network II (CAISN II).
Ontario Graduate Scholarship.

Muskoka Summit for the Environment.
Queen's University Summer Work Experience Program (SWEP).

## Competing Interests

The authors declare there are no competing interests.

## Author Contributions

- Sarah S. Hasnain conceived and designed the experiments, performed the experiments, analyzed the data, prepared figures and/or tables, authored or reviewed drafts of the article, and approved the final draft.
- Shelley E. Arnott conceived and designed the experiments, authored or reviewed drafts of the article, and approved the final draft.

## Data Availability

   The code and data are available in the Supplementary Files.

## Supplemental Information

Supplemental information for this article can be found online at http://dx.doi.org/10.7717/peerj.18608#supplemental-information.

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
