# Peer review of "Variation in behaviour of native prey mediates the impact of an invasive species on plankton communities"

_PeerJ, doi:10.7717/peerj.18608_

## Round 0.1 · original submission · Major Revisions

Dear Dr. Hasnain,

After this first review round the reviewers were positive regarding your manuscript. Still, some issues were raised require further improvements in a next version of the study. The main issues raised by the reviewers concern:

1) language improvements (the services of a third-party company are recommended to edit and proofread the text. Several companies are available elsewhere, so please take advantage of that).
2) mention more about dissolved oxygen concentration and environmental differences.
3) discuss the traits of the Daphnia species from different environments.
4) discuss/inform about stock density of the species.
5) Discuss the clarity regarding trait variation in mesocosms and population of analyzed species.
6) Further discussions on the role played by source populations (or the lack of it) on Daphnia trait variation.
7) Further discussions involving the effects of source locations on Daphnia vertical position and the lack of change in Daphnia vertical position caused by Bythotrephes-induced vertical response.

Considering all issues raised by the reviewers, I will grant you a major review.

Sincerely,
Daniel Silva

Reviewer 1 ·

Basic reporting

My main observations concern writing, which may benefit from a review. Besides that, the article is very well thought of. Ideas and design are coherent and detailed.

 Abstract (152 words and 1159 characters with spaces) needs review. The information is there, but can be better explained. See below an example of a review.
“However, the role of trait variation in prey populations on predator impacts at larger ecological scales is not well understood, especially for introduced predators where this variation can shape invasion outcomes.” (Lines 30-32)
“However, the impacts exerted on predatores by the variation of the traits of prey populations at larger ecological scales is not well understood, especially considering introduced predators, where this variation can shape invasion outcomes.”
Although longer in terms of wordcount, the idea becomes clearer in the reviewed version. There are several instances where the abstract will benefit from such review.
 Line 47: change “on prey trait expression” to “on the expression of prey traits”
 Line 57: with a species or within a species?
 Lines 72-73: This sentence seems a bit out of place, concerning the structure of the introduction. I suggest adding a sentence on the community level impact of predataion on prey communities and short statement on exotic species, before narrowing your reasoning to “non-native predators”
 Line 78: Please standardize the terms specific or population. I suggest using intra and inter population throughout the text.
 Line 79: I like the use of the word naivetè, but am concerned that it may affect the understanding negatively. I suggest using an english language synonim.
 Lines 83-88: Confusing sentence. Please rewrite.
 Line 89: Please specify. “Bythotrephes is a voracious zooplanktivore… (cladocera, microcrustacean, invertebrate, organism…?)
 Lines 95-96: Suggestion. Change to “induce trophic cascades and lead to increased phytoplankton biomass in invaded lakes”
 Line 105, 116, 125, 137, 139, 140, 145, 153, 394-395, 405, 408, 416, 417, 424, 430,459, 462, 464, 470, 475-76: change to “the vertical position of Daphnia”
 Line 112: change to i.e., (italics)
 Line 115: Why do you expect increases in only calanoid copepods? I would expect in cyclopoid as well.
 Line 153: please add a reference or some evidence in supplementary information
 Line 158: review sentence
 Line 240-241: “density of each”
 Lines 441-443: Please remove the double negative.

Experimental design

Very well designed and described. I have only one doubt. Was each layer sampled consecutively? If so. How did you control the effect of the net disrupting the layer before it was sampled?

Validity of the findings

The results and discussions are complete and in line with the experimental findings. The article was very well designed and developed

Additional comments

no comment

·

Basic reporting

Figure 2 seems to need the caption changed, as currently the description text only goes up to Fig. 2c. Any text relating to Fig. 2d is missing.

To correct/clarify the language please make the following changes:
Line 37: change to "and larger reductions in density were observed".
Line 57: is this meant to be "within a species"?
Line 58: I think this could be changed to "large spatial and temporal environmental heterogeneity".
Line 158: Delete "was".
Line 160: Change to "included in our analyses".
Line 198-199: delete repeated word.
Line 240-241: Is this meant to be "density of a zooplankton"?
Line 331: Please italicize the second part of the species name.
Line 363: Delete "in".

Experimental design

Line 131: Please make some mention about how the dissolved oxygen concentration might gradually start to differ between the water in the bags and the water in the wider lake. This is because the ratio of surface area to water volume must be lower for the water in bags compared to the rest of the lake. And after mesh has been put across the top of the bag (line 136) the oxygen exchange from air to water will be further reduced.

Line 153: Were the Daphnia from the two different lakes similar in most/all other traits, except the trait of tendency toward different vertical position?

Line 163: Was some field data used to determine this stocking density? (i.e. does 10 per m3 represent the density encountered naturally in the field?).

Line 239: some of the variables are defined by the conditions at week 3 (e.g. line 238) and other values by conditions at week 0 (e.g. line 239). If the total hypolimnetic Daphnia did not change from week 0 to week 3 (line 221-222) then I can understand just using the data from week 0. But if, as stated line 444, there were "observed changes in total Daphnia daytime vertical position in our mesocosms", then I would think it would make more sense to use the proportion of total hypolimnetic Daphnia from week 3 (i.e. after the changes have happened), or even some average position from throughout the experimental time. Overall, I think its important that the dynamics of the vertical position of the experimental Daphnia are made clear in the manuscript, so I ask that the data and discussion about this which is currently in the supplementary materials (e.g. as stated line 170) be added to the main manuscript text, and please clarify about the two seemingly contradicting statements from line 221 and line 444.

Validity of the findings

no comment

Additional comments

Line 70-71: It seems a bit obvious statement, that "prey trait variation impacts predator-prey dynamics". Whether using these kinds of terminology or not, surely the fact that all the kinds of prey trait variation can impact predator-prey dynamics is something that has been shown in very many laboratory and field studies, and also often assessed in a community context.

Line 115: But would not the invasive predator also eat the copepod? If the predator for some reason prefers to eat the copepod, then you may expect decreases in copepod abundance in invaded mesocosms regardless of Daphnia vertical position.

Reviewer 3 ·

Basic reporting

The manuscript is clearly written in unambiguous, professional language. It is well-referenced and provides sufficient background and context for the reader to understand the relevance of the work.

Experimental design

The research falls within the scope of the journal. The research question, 'Do differences in Daphnia vertical position influence impacts of Bythotrephes on zooplankton and primary production is well defined and it is clearly stated how this specific research fills the indentified knowledge gap on 'the role of prey trait variation on the impacts of invasive predators at large ecological scales'. The experimental and statistical approach exhibit rigor and methods are described with enough detail and clarity to repeat the experiment. There is one particular aspect of the experimental/statistical design that is not sufficiently accounted for, and that is the effect of source location on the findings. Daphnia were sourced from two locations (Echo Lake and Bonnie Lake) specifically to provide population-level trait variation in Daphnia vertical position. However, in the experiment, there were no a-priori differences in the proportion of hypolimnetic Daphnia in any of the mesocosms, and this appears to be the case regardless of Bythotrephes addition or source location. So there is something about the experiment (e.g., stress from handling, acclimation to novel mesocosm conditions, etc.) that equalized trait variation between the source locations (despite exceptionally large differences in the field prior to the experiment). The statistical analyses relies on mesocosm-level varation in Daphnia depth to make the central point that trait variation influences predator effects, but it is not clear whether the trait variation observed in the meoscosms is in any way related to source population (which is assumed in the manuscript to be the source of trait variation). I think the central finding of the manuscript remains intact (see section 3 below), but I think the manuscript could be improved by more explicitly discussing the role of source population (or lack thereof) on Daphnia trait variation in the experiment and more-so by explicitly including source population as a factor in the statistical models, which does not appear to have been done.

Another concern is that the volumetric density of Bythotrephes in the mesocosms was at the low end of what is observed in naturally invaded lakes. I'd like to see justification for the number of Bythotrephes that were used in the experiment, as it has bearing on why there was no observable effect of Bythotrephes on the vertical distribution of Daphnia in the mesocosms (see comments in Section 3 below).

Validity of the findings

According to the manuscript, there were no a priori differences in the proportion of hypolimnetic Daphnia between invaded and uninvaded mesocosms at the beginning of the experiment, and Daphnia daytime vertical position remained stable across mesocosms throughout the duration of the experiment. However, given that the natural populations that the Daphnia were sourced from exhibited huge differences in vertical positions, the reader is lead to believe that this population-level trait variation is what should be influencing Bythotrephes' direct and indirect predatory effects in the mesocosm. It seems however that the effect of source location on Daphnia vertical position in the mesocosms was not explicitly accounted for (see my comments in Section 2 above).

The manuscript also states that there were significant differences in the abundance of zooplankton between Echo and Bonnie lakes, which influenced initial zooplankton densities in the mesocosms - with
zooplankton density greater in enclosures stocked from Echo Lake as compared to Bonnie
Lake. This is important because if the trait variation in Daphnia is in part a plastic response to Bythotrephes presence (e.g., predator induced daytime vertical migration) then the abundance of conspecific and heterospecific zooplankton could modify this response. In fact the observation that Daphnia from Echo Lake, where zooplankton are much more abundant, exhibit a much shallower vertical distribution, is consistent with the theoretical prediction that Daphnia from Echo lake would respond less strongly to Bythotrephes presence because of the increase con- and heterospecific density in the source lake or mesocosm. This possibility also highlights the importance of considering the independent effects of source location in the statistical models, but also perhaps more importantly, the interactive effects of source location and Bythotrephes presence on Daphnia vertical position in the experiment.

Overall, it is concerning to me that (1) the effect of source location on Daphnia vertical position in the experiment was not explicitly considered, and (2) that Daphnia vertical position did not change during the experiment and specifically why there was no observable Bythotrephes-induced vertical response in Daphnia. Neither of these omissions diminish the conclusions that the manuscript provides about the impact of trait variation on predator impacts, but explicit consideration in the statistical models (in the case of source location effects) and in the Discussion (in the case of both source location effects and the lack of Bythotrephes-induced trait responses) would strengthen the manuscript by providing (or ruling out) some possible mechanisms for the sources of trait variation.

Additional comments

On Line 72, please clarify with ‘often’ as this is not always the case (see e.g., Bourdeau et al. 2013 Ecology).

In Lines 350-353 "Most notably, we found larger increases in phytoplankton
biomass in invaded mesocosms where there was little spatial overlap between Daphnia and
Bythotrephes, i.e., greater proportion of hypolimnetic Daphnia as compared to uninvaded
mesocosms." the sentence construction could be improved for clarity.

---

## Round 0.2 · Minor Revisions

Dear Dr. Hasnain,

After this new review round, both reviewers believe your manuscript is almost ready for acceptance. Please proceed with the final changes suggested by them and I also think we will be good to go.

Sincerely,
Daniel Silva

Reviewer 1 ·

Basic reporting

The authors made relevant changes to the manuscript and were very open to suggestions. I have no more comments.

Experimental design

No comment

Validity of the findings

No comment

Additional comments

no comments

·

Basic reporting

Here are a few corrects to the spelling/grammar:

Line 140: Please change "influences" to "influence".

Line 246: I think "use" needs to be deleted here.

Line 453: Please fix the sentence that ends here.

Line 481: This sentence seems to have an error in it.

Experimental design

no comment

Validity of the findings

Line 248-249: The statement here - "we observed changes in the proportion of total hypolimnetic Daphnia between week 0 and week 3" seems to directly contradict what was stated before at line 226-227 - "We found no significant differences in the proportion of hypolimnetic Daphnia between the beginning (week 0) and the end of the experiment (week 3". Please clarify about this in the Methods text.

Additional comments

Line 33-34: I think it would be more clear if it was mentioned here that the difference in vertical position, in this case, is the trait variation spoken about in the preceding sentences.

Line 137-139: It is confusing here that a statement is made which is based on "data", but then a citation is provided which is a personal communication. Is it not possible to get the data to show (e.g. in supplementary materials) from the personal communication?

---

## Round 0.3 · accepted · Accept

Dear Dr. Hasnain,

I am pleased to accept your manuscript for publication in PeerJ! Congratulations!

Sincerely,
Daniel Silva

Reviewer 1 ·

Basic reporting

The authors made all the changes requested by the reviewer. I have no further comments.

Experimental design

no comment

Validity of the findings

no comment

Additional comments

no comment